# The Impact of Facility Surgical Caseload Volumes on Survival Outcomes in Patients Undergoing Radical Cystectomy

**DOI:** 10.3390/cancers14235984

**Published:** 2022-12-03

**Authors:** Giovanni E. Cacciamani, Afsaneh Barzi, Michael B. Eppler, Primo N. Lara, Chong-Xian Pan, Sumeet K. Bhanvadia, Parkash Gill, Monish Aron, Inderbir Gill, Sarmad Sadeghi

**Affiliations:** 1Institute of Urology, USC Keck School of Medicine, Norris Comprehensive Cancer Center, University of Southern California, Los Angeles, CA 90033, USA; 2City of Hope Comprehensive Cancer Center, Department of Medical Oncology & Therapeutics Research, Duarte, CA 91010, USA; 3UC Davis Comprehensive Cancer Center, 4501 X Street, Sacramento, CA 95817, USA; 4Department of Medicine, Harvard Medical School, West Roxbury, MA 02132, USA; 5Department of Medicine, USC Norris Comprehensive Cancer Center, University of Southern California, Los Angeles, CA 90033, USA; 6Norris Cancer Hospital and Clinics Norris Comprehensive Cancer Center, University of Southern California, Los Angeles, CA 90033, USA

**Keywords:** bladder cancer, radical cystectomy, pelvic exenteration, oncologic outcomes, facility caseload, overall survival, NCDB, national cancer database

## Abstract

**Simple Summary:**

The oncological outcome of curative intent surgery for urothelial carcinoma of the bladder could be impacted by facility caseload and surgical experience. In this review of 27,272 cases of radical cystectomy and pelvic exenteration, as the annual caseload of surgery at the facility decreased, the all-cause mortality for the patients increased significantly. The caseload of the facility where radical cystectomy and pelvic exenteration were performed had a direct and significant impact on the overall survival for the patients.

**Abstract:**

The role of surgical experience and its impact on the survival requires further investigation. A cohort of patients undergoing radical cystectomy or anterior pelvic exenteration for localized bladder cancer between 2006 and 2013 at 1143 facilities across the United States was identified using the National Cancer Database and analyzed. Using overall survival (OS) as the primary outcome, the relationship between facility annual caseload (FAC) and facility annual surgical caseload (FASC) for those undergoing curative surgery was examined. Four volume groups (VG) depending on caseload using both FAC and FASC were defined. These included VG1: below 50th percentile, VG2: 50th–74th percentile, VG3: 75th–89th percentile, and VG4: 90th and above. Between 2006 and 2013, 27,272 patients underwent surgery for localized bladder cancer. The median OS was 59.66 months (95% CI: 57.79–61.77). OS improved significantly as caseload increased. The unadjusted median OS difference between VG1 and VG4 was 15.35 months (64.3 vs. 48.95 months, HR 1.19 95% CI: 1.13–1.25, *p* < 0.001) for FAC. This figure was 19.84 months (66.89 vs. 47.05 months, HR 1.25 95% CI: 1.18–1.32, *p* < 0.0001) for FASC. This analysis revealed a significant and clinically important survival advantage for curative bladder cancer surgery at highly experienced centers.

## 1. Introduction

The impact of experience on outcomes of curative intent surgery for bladder cancer has previously been investigated and reported [1,2,3,4,5,6,7,8,9,10]. This effect was also studied in prostate cancer among others [5,11,12,13,14]. While the association between volume and outcomes is well-established, the factors responsible for this remain elusive. Such factors could include the primary organ involved, patient characteristics or disease characteristics, or even adherence to standards of care and professional guidelines for cancer care.

Furthermore, it should be questioned whether experience should be defined by the individual surgeon or the cumulative experience at the facility where the intervention occurs. The goal of this study is to analyze the determinants of overall survival for bladder cancer surgery.

## 2. Materials and Methods

### 2.1. Data Source

The National Cancer Database (NCDB) includes many cancer case histories, including bladder cancer, treated in the United States at Commission on Cancer (CoC) accredited facilities. It contained treatment outcomes for bladder cancer at over 1231 facilities during 2004–2013.

### 2.2. Variables

A list of NCDB variables is available at https://www.facs.org/quality-programs/cancer/ncdb (access on 1 January 2016). Overall survival was used as the primary outcome, defined as time from the day of surgery to the date of death from any cause. This variable was censored on the date of last follow-up. Secondary outcomes included operative outcomes of surgical margin status and nodal status, as well as postoperative radiation therapy and/or chemotherapy.

The facility annual caseload (FAC) and facility annual surgical caseload (FASC) variables were derived and used to assign facilities to one of 4 volume groups (VG): (1) below 50th percentile, (2) 50–74th percentile, (3) 75–90th percentile, and (4) 90th+ percentile. Using the results of the pathology report after the surgical procedure and based on the National Cancer Comprehensive Network (NCCN) practice guidelines (whether neoadjuvant/adjuvant chemotherapy or adjuvant radiation was indicated or given) the appropriateness of postoperative management was determined. In addition to volume groups (VG), other variables such as patient demographics, disease characteristics, and facility characteristics were used in the multivariable analysis.

### 2.3. Patient Selection

Patients diagnosed with localized urothelial bladder cancer who underwent radical cystectomy or anterior pelvic exenteration for urothelial carcinoma were selected. Patients with metastatic disease and those who died within 90 days of surgery were excluded. (Figure 1).

### 2.4. Hypothesis

The hypothesis was that surgeon experience based on annual caseload was a predictor of oncological outcome. Facility annual caseload (FAC) and facility annual surgical caseload (FASC) were studied separately.

### 2.5. Statistical Analysis

Based on FAC and FASC, descriptive analysis was used to evaluate patient, disease, and facility characteristics stratified by volume group (VGs). Multivariable Cox proportional hazards models were calculated for the analysis of overall survival (OS) by volume group (VG) adjusting for prognostic factors. Using the robust sandwich estimates of Lin and Wei, the correlation of outcomes of patients treated in the same facility was addressed [15]. Sensitivity analyses were performed to examine the impacts of 90-day mortality and volume group.

All analyses were performed using Stata statistical software (version 15; Stata Corp., College Station, TX, USA). All tests were 2-sided with alpha of 0.05.

**Figure 1 cancers-14-05984-f001:**
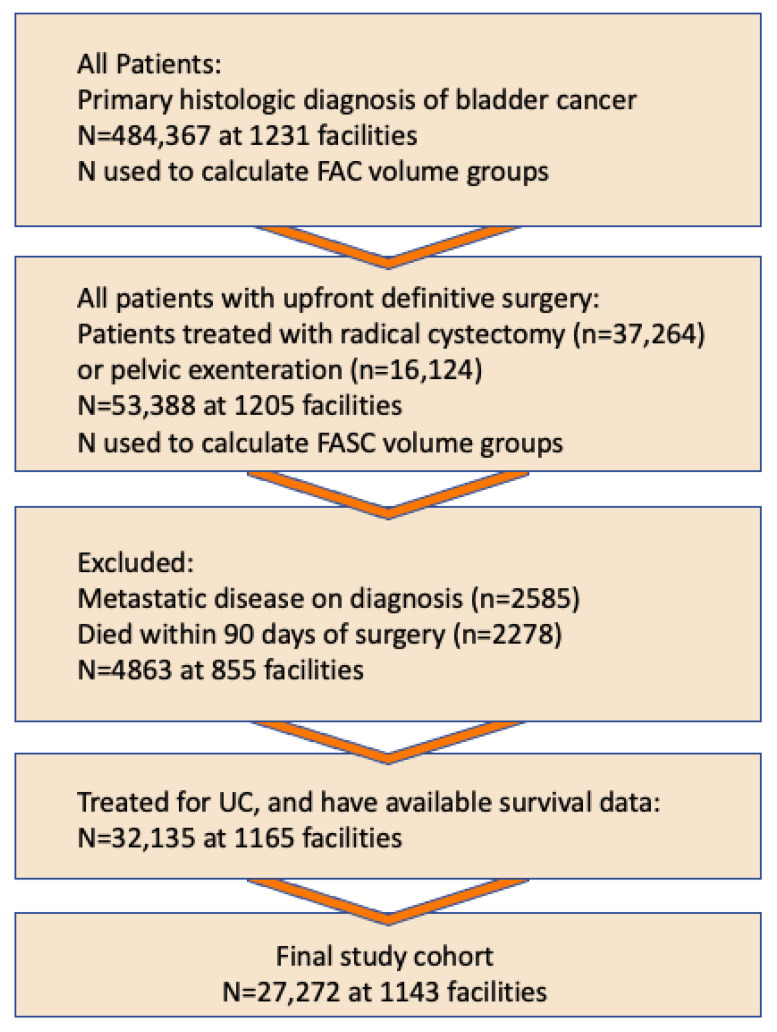
Cohort selection process. As the focus of this analysis and the hypothesis tested here is the impact of caseload on the oncologic outcome of bladder cancer surgery, the 90-day mortality, which is largely reflective of death due to surgical complications rather than progressive disease, is excluded from the analysis. Similarly, this analysis is focused on curative intent surgery for bladder cancer, and therefore, patients with known metastatic disease who underwent surgery primarily for palliative reasons are excluded.

## 3. Results

Between 2006 and 2013, 484,367 patients diagnosed with bladder cancer were treated at 1231 facilities in the United States. There were 27,272 patients who underwent radical cystectomy or pelvic exenteration at 1143 facilities as the primary therapy for localized urothelial carcinoma of the bladder. Median OS for the patients was 59.66 months (95% CI: 57.79–61.77) with interquartile range of 20.27 to 116.67 months. Over 91% of patients were of white race and 32.6% had private insurance. By treatment center, 52 and 32% of patients were treated at academic and comprehensive community cancer programs, respectively. The majority (70%) of patients had a comorbidity score of 0, while 7% had a comorbidity score of ≥2. Table 1 reports patient characteristics.

Approximately, 43% of these facilities were comprehensive community cancer programs, and 26% were community cancer programs. Using FASC—facility annual surgical caseload—886 facilities were below median (volume group 1), 713 in the 50–74th percentile (volume group 2), the 419 in 75–89th percentile (volume group 3), and 190 in the top 10 percentile (volume group 4). Using FAC—all bladder cancer annual caseload—689 facilities were below median, 596 in the 50–74th percentile, 387 in the 75–89th percentile, and 202 in the top 10 percentile. Table 2 summarizes the facility types and number of patients treated at each facility type over the study period. Of note, over the years, facilities could move from one VG to another, and therefore the numbers do not add up to 1143.

### 3.1. Results by Facility Volume Groups

Using facility annual surgical caseload (FACS), the numbers of radical cystectomies of 0–2, 3–5, 6–11, and > 11 reflected surgical volumes of less than the 50th, the 50–74th, the 75–89th and the top 10 percentiles, respectively. These figures were <0–28, 29–47, 48–70, and >70 based on facility annual volume (FAC), respectively. More than half of patients underwent surgery at academic/research programs, which make up 11% of facilities. The highest-volume facilities were mostly in academic/research programs, while most of the surgical procedures in VG1 were performed at comprehensive community cancer programs (Table 2).

An average of 11 lymph nodes were examined, with interquartile range of (IQ: 5–21) for all surgeries. However, the median and interquartile range were slightly higher as the surgical experience increased: VG1 8 (IQ: 2–17), VG2 8 (IQ: 3–15), VG3 9 (IQ: 4–16), VG4 15 (IQ: 8–24). The average node positivity rate was 4.9% (Appendix A).

### 3.2. Univariable and Multivariable Survival Analyses

In this population of 27,272 patients who underwent radical cystectomy or pelvic exenteration, the overall survival improved as annual caseload increased. In the univariable model for FASC, a 25% increased risk of mortality was detected when the highest- and lowest-volume groups were compared (HR of 1.25—95% CI: 1.18–1.33), *p* = 0.0001. For FAC, there was a 19% increased risk of mortality when VG4 was compared with VG1 with a HR of 1.19 (95% CI: 1.13–1.25), *p* = 0.0001.

When the model was adjusted for perioperative chemotherapy, age, gender, education, comorbidities, number of nodes examined, and clinical and pathologic node status, the hazard ratios were smaller but still significant between volume groups 1 and 4. For both FASC and FAC, there was a 12% increased risk of mortality when VG4 was compared with VG1 with a HR of 1.19 (95% CI: 1.09–1.14), *p* = 0.0001—Figure 1.

Under the FASC model, based on perioperative chemotherapy and whether neoadjuvant and adjuvant chemotherapies were indicated and given, patients were divided into multiple subgroups. Those who met the indication for neoadjuvant chemotherapy and were given such chemo were considered the reference group (subgroup 12).

The outcomes of patients who met the neoadjuvant chemotherapy indication but were not given such chemo and received chemo in adjuvant setting even though pathology specimen did not support the use of adjuvant chemotherapy (subgroup 9) were not statistically significantly different from the reference group (Figure 1).

Patients who met the indication for neoadjuvant chemo and adjuvant chemo but were not given either (subgroup 10) and patients who met the indication for adjuvant but did not receive chemotherapy (subgroup 2) had the worst outcomes, with HR of 2.74 (95% CI: 2.46–3.05) and 2.63 (95% CI: 2.35–2.94), respectively.

Lymph node dissection with the examination of more than 30 nodes was associated with better outcomes, with HR of 0.55 (95% CI: 0.51–0.60) (Figure 1). The FAC model results were similar (Figure 2). Other variables such as patients’ insurance type and geographic location did not have a significant impact on outcomes.

### 3.3. Overall Survival Curves

When the highest- and lowest-volume groups, i.e., VG4 and VG1, were compared, the OS advantage reached 19.8 months for FASC and 15.4 months for FAC. The difference for adjusted median OS comparing VG4 with VG1 was similar at 10.8 months for both FASC and FAC models (Figure 3a–d).

**Figure 2 cancers-14-05984-f002:**
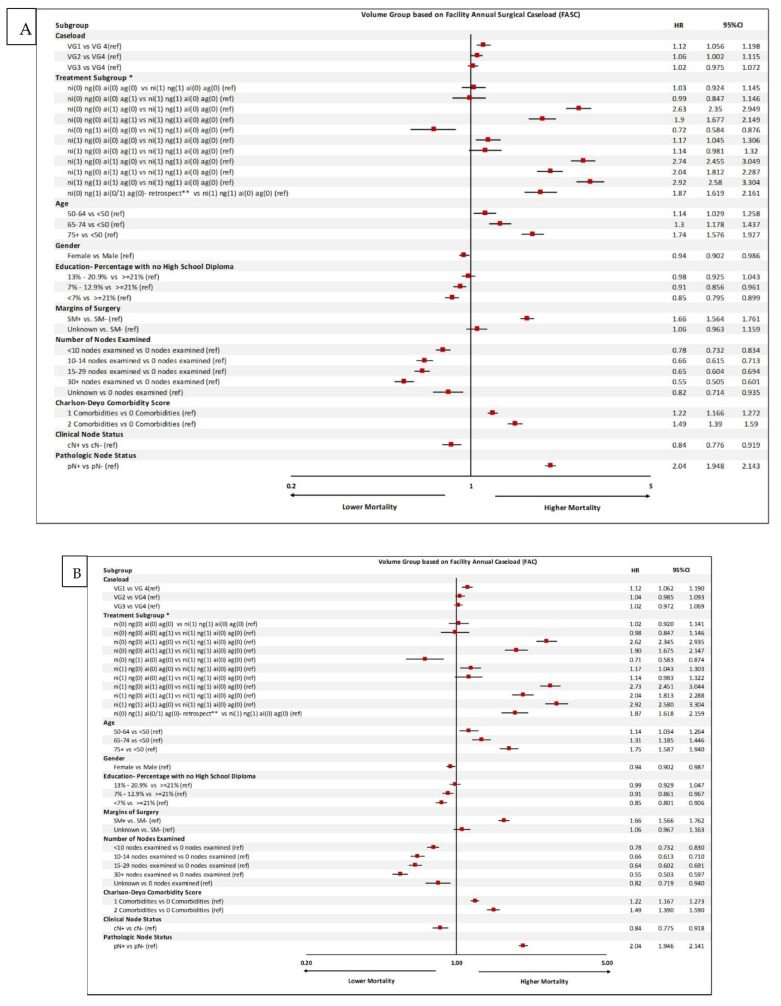
Multivariable logistic regression analysis of predictors of mortality based on (**A**) facility annual surgical caseload (FASC) and (**B**) facility annual caseload (FAC). HR: hazard ratio; SM: surgical margin; VG: volume group; * refers to the combination of the following: NI: neoadjuvant chemo indicated; NG: neoadjuvant chemo given; AI: adjuvant chemo indicated; AG: adjuvant chemo given. ** These are patients who received neoadjuvant chemo without a clear indication, but post operatively based on pathology findings would have qualified for adjuvant chemo. (0) no, (1) yes.

## 4. Discussion

This analysis revealed an overall survival benefit favoring performing radical cystectomy at facilities with higher annual caseloads. The all-cause mortality risk for the unadjusted FASC model was 25% lower and favored the highest-volume centers when compared with the lowest end of the volume. When the FASC model was adjusted for other factors, the difference was still 12%. This was associated with a large 10.8-month survival difference at the median when VG4 (highest-volume group) was compared with VG1 (lowest-volume group). The FASC model is a better predictor of outcomes than the FAC model in the unadjusted analysis. However, in the adjusted multivariable analysis, the FASC and FAC models were equally good predictors of outcomes. This can be attributed to the significance of other interdisciplinary factors involved in the care of patients that impact the outcome beyond the individual surgeon’s experience.

Only the highest-volume facilities (either by FASC or FAC) had a higher-than-average negative surgical margin rate. The average number of nodes examined was higher at high-volume centers by either method, although slightly higher by FASC vs. FAC model. The higher number of nodes examined was also a predictor of better overall survival. The Charlson–Deyo comorbidities index had a small effect on the outcome differences among volume groups. Case-mix variations across volume groups also did not have sufficient impact to account for the outcome differences.

Despite the inclusion of various prognostic factors in the multivariable model, the impact of annual caseload on the outcomes remained highly significant. However, the separation between VG 2, 4, and 4 decreased; under the adjusted FASC model, VG 3 and 4 were not statistically significantly different, and under FASC, VG 2, 3, and 4 were not statistically significantly different. This provides the interesting insight that facilities with 6 or more radical cystectomies annually or 29 or new bladder cancer cases annually may be providing optimal care in terms of average outcomes.

A similar study of radical cystectomies in Canada evaluated 3296 patients and estimated small but statistically significant unadjusted and adjusted survival benefits for performing radical cystectomy at high-volume centers with HR 0.994, *p* = 0.014 and 0.995, *p* = 0.044, respectively [2]. A Dutch population-based study of 2168 patients reported a survival advantage for performing radical cystectomy at high-volume centers by looking at facilities with less than 10 cystectomies vs. 10 or more cystectomies per year, unadjusted HR 1.21, adjusted HR 1.17, *p* < 0.05 [16]. A SEER-Medicare study of 7127 patients looked at hospital volume and surgeon volume to determine which one is more critical and concluded that the hospital volume may be more important than the individual surgeon’s experience [3].

Several series suggest that early hospital outcomes may drive overall survival benefits and impacts of case volume and early survival outcomes have been reported previously. In prior work, the association of hospital volume with conditional 90-day mortality after cystectomy was examined. The investigators reported that the 30-day and 90-day mortality rates were lower for high-volume hospitals when compared with low-volume hospitals [17]. Of note, the definition for hospital volume was arbitrary, and the short follow-up considered did not allow for appreciating the cancer-specific survival of patients undergoing RC [17].

In an analysis of radical cystectomy using NCDB surgical outcomes and survival data, the investigators noted that surgical outcomes and survival improved with high hospital volume and academic status [9]. Our findings, with a mean follow-up of 32.5 months, highlight an important aspect of surgical care and contribute to the already robust data on volume–outcome relationships in urologic surgery. Moreover, we compared the surgical volumes adjusted for FAC with FACS. Of note, we provide an analysis-based definition of surgical volume. The present study benefits from a much larger sample size, includes the effects of adjuvant and neoadjuvant chemotherapy, has a more granular view of the volume, and instead of looking at pure surgical experience, takes a broader view of experience with all stages of bladder cancer for the outcomes.

Limitations of this study include the fact that it is a retrospective analysis of a dataset from a registry that includes only all-cause mortality rather than cancer-specific mortality. The variables in a registry are incomplete in nature, do not provide a full picture, and have significant missing data. Furthermore, facility selection biases are not captured by this type of dataset and cannot be modeled. Invariably, the treatment decisions are based on pathology and imaging reports, and outcomes can vary significantly based on the quality of such reports. These variables could not be modeled based on the NCDB dataset.

## 5. Conclusions

There is a clinically important and statistically significant survival advantage for performing radical cystectomies at highly experienced centers based on the analysis presented here.

## Figures and Tables

**Figure 3 cancers-14-05984-f003:**
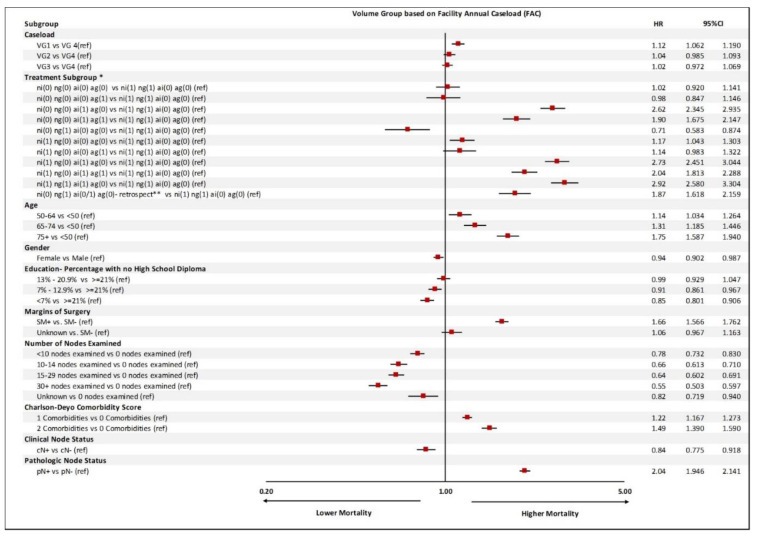
Survival curves adjusted by caseload volumes. FAC: facility annual caseload; FASC: facility annual surgical caseload. HR: hazard ratio; SM: surgical margin; VG: volume group; * refers to the combination of the following: NI: neoadjuvant chemo indicated; NG: neoadjuvant chemo given; AI: adjuvant chemo indicated; AG: adjuvant chemo given. ** These are patients who received neo-adjuvant chemo without a clear indication, but post operatively based on pathology findings would have qualified for adjuvant chemo. (0) no, (1) yes.

**Table 1 cancers-14-05984-t001:** Patient characteristics.

Variable	N	%
Number of Patients	27,272	
Median Age at Diagnosis	68	
Interquartile range	(61, 75)	
Range	(21, 90)	
Race/ethnicity		
White	24,883	91.2%
Black	1508	5.5%
Other	587	2%
Primary Payor		
Not Insured	800	2.9%
Private Insurance	8897	32.6%
Medicaid	1144	4.2%
Medicare	15,642	57.4%
Other Government	271	1.0%
Insurance Status Unknown	518	1.9%
Charlson–Deyo Comorbidity Score		
0	19,263	70.6%
1	6146	22.5%
2	1863	6.8%
Great Circle Distance		
Median	16	
Facility Type		
Community Cancer Program	1727	6.3%
Comprehensive Community Cancer Program	8705	31.9%
Academic/Research Program	14,139	51.8%
Integrated Network Cancer Program	2535	9.3%

**Table 2 cancers-14-05984-t002:** Facility characteristics and distribution of radical cystectomies based on volume group.

Volume Group Based on Facility Annual Surgical Caseload (FASC)								
Volume Group	1		2		3		4	
Percentile [volume range]	<50th [0–2]		50–74th [3–5]		75–89th [6–11]		90th+ [12+]	
Number of Facilities (total)	903	100%	748	100%	464	100%	251	100%
Community Cancer Program	314	35%	133	18%	25	5%	4	2%
Comprehensive Community Cancer Program	462	51%	417	56%	235	51%	54	22%
Academic/Research Program	98	11%	137	18%	133	29%	111	44%
Integrated Network Cancer Program	19	2%	36	5%	41	9%	26	10%
Unknown	10	1%	25	3%	30	6%	56	22%
Unique Facility Counts *	886		713		419		190	
Number of Patients Served (total)	2804	100%	4584	100%	6355	100%	13,529	100%
Community Cancer Program	984	35%	586	13%	142	2%	15	0%
Comprehensive Community Cancer Program	1506	54%	2795	61%	3303	52%	1101	8%
Academic/Research Program	266	9%	932	20%	2175	34%	10,766	80%
Integrated Network Cancer Program	38	1%	244	5%	702	11%	1551	11%
Unknown	10	0%	27	1%	33	1%	96	1%
Volume Group Based on Facility Annual Caseload (FAC)								
Volume Group	1		2		3		4	
Percentile [volume range]	<50th [0–28]		50–74th [29–47]		75–89th [48–70]		90th+ [71+]	
Number of Facilities (total)	717	100%	633	100%	423	100%	257	100%
Community Cancer Program	308	43%	83	13%	10	2%	7	3%
Comprehensive Community Cancer Program	293	41%	394	62%	241	57%	77	30%
Academic/Research Program	82	11%	112	18%	114	27%	92	36%
Integrated Network Cancer Program	9	1%	13	2%	30	7%	40	16%
Unknown	25	3%	31	5%	28	7%	41	16%
Unique Facility Counts *	689		596		387		202	
Number of Patients Served (total)	3828	100%	5011	100%	6611	100%	11,822	100%
Community Cancer Program	1321	35%	323	6%	48	1%	35	0%
Comprehensive Community Cancer Program	1653	43%	2950	59%	2613	40%	1489	13%
Academic/Research Program	794	21%	1587	32%	3441	52%	8317	70%
Integrated Network Cancer Program	33	1%	119	2%	475	7%	1908	16%
Unknown	27	1%	32	1%	34	1%	73	1%

* Because a facility may have a change in type designation or volume group memberships the breakdown sums do not reflect the unique facility numbers as shown in this column.

## Data Availability

Restrictions apply to the availability of these data. Data were ob-tained from the National Cancer Database and are available from the authors with the permission of the National Cancer Database.

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
