# Peer review of "The Impact of Facility Surgical Caseload Volumes on Survival Outcomes in Patients Undergoing Radical Cystectomy"

_cancers, 2022, doi:10.3390/cancers14235984_

Round 1

Reviewer 1 Report

The authors report on the impact of surgical volume on outcome after radical cystctomy, based on a very large nation-wide dataset. The major strength of this study in addition to the very large sample size is the inclusion of other parameters relevent with respect to OS, like perioperative chemotherapy, histopathological parameters and comorbidities, enableing a robust multivariable analysis.

Comments:

Please briefly define FAC and the difference to FASC. If FAC represents all bladder cancer patients and the difference refers to bladder cancer patients ultimately not undergoing cystectomy, why is the number of cases in the volume group 4 higher for FASC than for FAC?

Patient selection:

The drop-out rate is high. Out of 53388 patients undergoing cystectomy, only 51% were available for analysis. Apparently, lack of survival data was the main reason for exclusion. It sould be discussed whether this drop out rate represents a possible bias.

The exclusion of patients who died within 90 days of surgery needs to be clarified. 90-day mortality, affecting 7% of patients in this series, is predominantly a result of surgical complications, which in turn are usually related to caseload. On the other hand, mortality later than 90 days is more frequently caused by cancer progression or comorbid conditions, which represent parameters less closely related to the quality of surgery. Since the article is intended to evaluate the impact of caseload on overall survival, exclusion of 90 day mortality results in a selection bias. Thus, these patients should be included.

Exclusion of metastatic patients: It makes sense to exclude patients with distant metastastic disease undergoing surgery for palliative reasons. However, the number of almost 2600 patients (8% of the whole cohort) undergoing cystectomy with metastatic disease is surprisingly high and deserves a brief discussion.

References:

Ref. #7 is incomplete

Several other studies have reported on hospital and/or surgeon volumes and outcomes. Many of these are not cited, but at least the review article by Gossens-Laan et al, Eur Urol 2011, should be cited and discussed.

Author Response

Rev 1

The authors report on the impact of surgical volume on outcome after radical cystctomy, based on a very large nation-wide dataset. The major strength of this study in addition to the very large sample size is the inclusion of other parameters relevent with respect to OS, like perioperative chemotherapy, histopathological parameters and comorbidities, enableing a robust multivariable analysis.

Comments:

  1. Please briefly define FAC and the difference to FASC. If FAC represents all bladder cancer patients and the difference refers to bladder cancer patients ultimately not undergoing cystectomy, why is the number of cases in the volume group 4 higher for FASC than for FAC?

Reply#1 We thank the reviewer. The difference between FAC and FACS is highlighted in the methods section.

  1. Patient selection:

The drop-out rate is high. Out of 53388 patients undergoing cystectomy, only 51% were available for analysis. Apparently, lack of survival data was the main reason for exclusion. It sould be discussed whether this drop out rate represents a possible bias.

Reply#2 We thank the reviewer for the comment. This should not represent any form of systematic bias. We had no role in collection of this data by NCDB. As our outcome of interest were oncologic in nature and consistent with widely accepted endpoint such as overall survival, including those who have such data available seems reasonable and would not affect any subgroups examined here in a more favorable or unfavorable light. As this is done at the level of screening for availability of datapoints of interest, the authors don’t feel this requires a specific paragraph in the manuscript. However, if our esteemed reviewer colleague feels otherwise, we are happy to add the statement here to the manuscript. 

  1. The exclusion of patients who died within 90 days of surgery needs to be clarified. 90-day mortality, affecting 7% of patients in this series, is predominantly a result of surgical complications, which in turn are usually related to caseload. On the other hand, mortality later than 90 days is more frequently caused by cancer progression or comorbid conditions, which represent parameters less closely related to the quality of surgery. Since the article is intended to evaluate the impact of caseload on overall survival, exclusion of 90 day mortality results in a selection bias. Thus, these patients should be included.

Replt#3.We thank the reviewer for the excellent point. The primary goal of this analysis is to address the effect of caseload on the oncologic outcome of the patients. We refer to the hypothesis. Therefore, post operative care and skills that impact the 90 day survival are not of the primary interest here. It stands to reason to believe higher level of caseload can impact both endpoints in the same fashion and direction. That is to say, both 90 day survival and oncologic survival will improve. However, as this is not core to the hypothesis, we did not examine I directly during the analysis. This will further investigated in a  a future study by our team.

  1. Exclusion of metastatic patients: It makes sense to exclude patients with distant metastastic disease undergoing surgery for palliative reasons. However, the number of almost 2600 patients (8% of the whole cohort) undergoing cystectomy with metastatic disease is surprisingly high and deserves a brief discussion.

Reply#4. We are happy to add to the caption of Figure 1 an explanation for the 90 day mortality exclusion and metastatic disease exclusion.

“Figure 1: Cohort selection process. As the focus of this analysis and the hypothesis tested here is on the impact of caseload on the oncologic outcome of bladder cancer surgery, the 90-day mortality which is largely reflective of death due to surgical complications rather than progressive disease is excluded from the analysis. Similarly, this analysis is focused on curative intent surgery for bladder cancer and therefore, patients with known metastatic disease who undergo surgery primarily for palliative reasons are excluded.”

References:

  1. Ref. #7 is incomplete

Reply#5  We thank the reviewer. We have changed ref# 7

  1. Several other studies have reported on hospital and/or surgeon volumes and outcomes. Many of these are not cited, but at least the review article by Gossens-Laan et al, Eur Urol 2011, should be cited and discussed.

Reply#6. We thank the reviewer. We have changed ref# 7 and we have added Gossens-Laan et al, Eur Urol 2011, which should be cited and discussed as suggested by the esteemed reviewer.

Reviewer 2 Report

Q1: the authors present a retrospective series of 27272 cases, and compare the volume and overall survival for bladder cancer surgery. I’m somewhat unsure about the innovation, and it’s obvious that survival advantage at highly experienced centers. Moreover, because only those with bladder cancer are included, it is not appropriate to represent facility surgical caseload volumes. Besides the FASC, it’s better to choose annual surgical caseload (including various urological surgery, more than radical cystectomy or pelvic exenteration) instead of FAC. Because the caseload of operation for benign diseases and other cancers better represent the facility experience.

Q2: Patients with metastatic disease and those who died within 90 days of surgery were excluded. My main concern is the study design. I wondered why you exclude the patients died within 90 days of surgery, and it would have been so much relevant to outcome. This might induce a selection bias. 

Q3: How authors distinguish higher volume groups from lower? The FAC and FASC variables were derived and used to assign facilities to one of 4 Volume Groups (VG): 1) below 50th percentile, 2) 50-74th percentile, 3) 75-90th percentile, and 4) 90+th percentile. How they were established in this study? The volume range of VG1 is 0-2, and VG2 is 3-5 based on FASC, which is no essential difference.

Q4: There are typos and errors in the manuscript (such as Lines 18, 139, 147, 153). There is no description of each subgroup (Figure 2). References must be written according the instructions of the journal.

Author Response

Q1: the authors present a retrospective series of 27272 cases, and compare the volume and overall survival for bladder cancer surgery. I’m somewhat unsure about the innovation, and it’s obvious that survival advantage at highly experienced centers. Moreover, because only those with bladder cancer are included, it is not appropriate to represent facility surgical caseload volumes. Besides the FASC, it’s better to choose annual surgical caseload (including various urological surgery, more than radical cystectomy or pelvic exenteration) instead of FAC. Because the caseload of operation for benign diseases and other cancers better represent the facility experience.

Reply#. We appreciate the reviewer’s viewpoint. It does stand to reason to think that more experience centers and surgeons will have better outcomes. But this assumption does not help us delineate what constitutes a high-volume center or surgeon. This is one of the questions we try to answer here. Furthermore, various studies have suggested volume cut offs that result in a change in outcome, mostly in prostate cancer. Then again, we are looking at this question in terms of facility and not individual surgeon. Once we take this approach, it immediately follows that we should also explore whether it is the individual surgeon or the center that makes the difference. It is possible that the collective efforts of the center that would include medical oncology expertise, radiation oncology expertise, and pathology expertise may be equally or more important. This is another question we try to answer. Finally, oncologic surgery has become and an entity of its own and in most high-volume centers oncologic urologists rarely do benign cases. Therefore, an examination of oncologic volume seemed more reasonable. As a side note, we would also like to add that NCDB does not include information about volume of benign cases and, therefore, we would not have access to such data to further test this hypothesis whether it is total urologic surgery volume, oncologic urologic surgery volume or bladder cancer surgery volume that matters. 

We hope our esteemed colleague finds this response adequate.  

Q2: Patients with metastatic disease and those who died within 90 days of surgery were excluded. My main concern is the study design. I wondered why you exclude the patients died within 90 days of surgery, and it would have been so much relevant to outcome. This might induce a selection bias. 

Reply#2. Thanks to the reviewer for the comment. The primary goal of this analysis is to address the effect of caseload on the oncologic outcome of the patients. We refer to the hypothesis. Therefore, post operative care and skills that impact the 90 day survival are not of the primary interest here. It stands to reason to believe higher level of caseload can impact both endpoints in the same fashion and direction. That is to say, both 90 day survival and oncologic survival will improve. However, as this is not core to the hypothesis, we did not examine I directly during the analysis.  This will further investigated in a  future study by our team.

Q3: How authors distinguish higher volume groups from lower? The FAC and FASC variables were derived and used to assign facilities to one of 4 Volume Groups (VG): 1) below 50th percentile, 2) 50-74th percentile, 3) 75-90th percentile, and 4) 90+th percentile. How they were established in this study? The volume range of VG1 is 0-2, and VG2 is 3-5 based on FASC, which is no essential difference.

Reply#3. Volume group membership of a facility is determined by the number of bladder cancers they treat (FAC) or bladder surgeries they perform (FASC). All facilities are lines up based on the volume and then depending on which percentile they belong to, they are placed in a volume group. 

The observation that doing 0-2 surgeries (FACS) represents a low volume and so does 3-5 bladder cancer cases per year (FAC), is very accurate. But the two measures are very different. In higher volume ranges, some facilities may be in FAC VG 4 but FASC VG3 or even VG2. This classification allows us to determine whether it is pure surgical skills or a multidisciplinary effect that accounts for some of the differences in outcomes.  We expect the low volume centers to be similar. That is why the group is large (below 50thpercentile). This distinction comes into forefront as volumes go up. 

Q4: There are typos and errors in the manuscript (such as Lines 18, 139, 147, 153). There is no description of each subgroup (Figure 2). References must be written according to the instructions of the journal.

Reply#4. We thank the reviewer for the comment. We have edited the figure 2 description and we have edited the references accordingly to Cancers

Round 2

Reviewer 1 Report

The authors clarified the exclusion of 90 day mortality, since the primary goal was to evaluate the impact of caseload on oncologic outcomes. Consequently, the title of the article should be modified and "survival" replaced by "oncologic outcomes". Otherwise, the title is misleading. All other aspects have been implented sufficiently.

Author Response

We thank the reviewer for the suggestion. It is our opinion that “survival” reflect better the outcome of interest of the present study as previously published for consistency (Barzi A, Lara PN Jr, Tsao-Wei D, et al. Influence of the facility caseload on the subsequent survival of men with localized prostate cancer undergoing radical prostatectomy. Cancer. 2019;125(21):3853-3863. doi:10.1002/cncr.32290” )